# Characterization of Tissue Equivalent Materials Using 3D Printing for Patient-Specific DQA in Radiation Therapy

**Yona Choi** [1,†] , **Young Jae Jang** [1,2,†] , **Kum Bae Kim** [2], **Jungbae Bahng** [1,*] and **Sang Hyoun Choi** [2,*]

1 Department of Accelerator Science, Korea University Sejong Campus, Sejong 30019, Korea
2 Research Team of Radiological Physics & Engineering, Korea Institute of Radiological and Medical Sciences, Seoul 139-706, Korea
* Correspondence: bahngjb@korea.ac.kr (J.B.); shchoi@kirams.re.kr (S.H.C.)
† These authors contributed equally to this work.

**Abstract:** Three-dimensional printing technology has the advantage of facilitating the construction of complex three-dimensional shapes. For this reason, it is widely used in medical and radiological fields. However, few materials with high electron density similar to that of bone exist for fabricating a human phantom. In this study, commercially available filament materials were used with an FDM 3D printer to perform delivery quality assurance (DQA) and were evaluated for medical use. For the bone filament material, $BaSO_4$ was synthesized in five ratios of 2%, 4%, 6%, 8%, and 10% with 40% PBAT and 50~58% PLA. The electron density for the 3D printing material fabricated was obtained using kV energy CT and compared with the electron density of human organs and bones. The radiation beam properties of the 3D printed structures were analyzed as films for treatment using a linear accelerator. As a result, by changing the infill density of the material, it was possible to produce a material similar to the density of human organs, and a homogeneous bone material with HU values ranging from $371 \pm 9$ to $1013 \pm 28$ was produced. The 3D printing material developed in this study is expected to be usefully applied to the development of a patient-specific phantom to evaluate the accuracy of radiotherapy.

**Keywords:** 3D printing technology; fused deposition modeling (FDM); tissue equivalent materials; delivery quality assurance (DQA); computed tomography (CT); linear accelerator

## 1. Introduction

With the technological advancements in 3D printing in recent years, various studies are being carried out. In particular, various studies have been conducted in the field of radiation therapy to evaluate the quality of medical 3D printing and to verify the physical and biological characteristics of radiation treatment by creating a customized shape for each patient [1–4].

A commercially available phantom with a density similar to that of the human body can be manufactured and compared based on medical factors for each patient using a 3D printer, and it can also be used as an appropriate tool to verify treatment plans through delivery quality assurance (DQA) [5–7].

Since the CT images used for treatment planning in radiation therapy are mainly acquired by using energy of 50 keV or more, the interaction with radiation is mainly due to Compton scattering and is closely related to the electron density of organs, bones, and tumors [8]. For human tissue materials, data on the electron beam attenuation coefficient can be represented by a single straight line, but for materials such as bone, the attenuation coefficient is different [9].

Currently, commercially available phantoms have disadvantages in terms of high cost, difficulty in realizing the physical properties desired by the user, and the long processing time of the material. In addition, although phantom manufacturers provide standard

values for several reference materials, research on materials with various densities for each organ, considering the different characteristics of each individual patient, is necessary [10]. Various studies have been conducted using various methods in the fields of medicine and medical physics using 3D printing technology [11–14].

Three of the most common 3D printing technologies are stereolithography (SLA), selective laser sintering (SLS), and fused deposition modeling (FDM).

SLA uses a photopolymer resin that is cured to form a model layer using an ultraviolet laser. It is possible to create models with various characteristics, such as strength and flexibility, and to fabricate precise models. SLS uses a mixed powder that is cured to form layers using a high-power laser. It can be used layer-by-layer without supports between the layers, and a wider variety of plastics, metals, or glass can be used to form 3D models [15]. These two printing techniques have the advantages of a high printing speed and precise printing; however, they also have disadvantages owing to their very high starting cost and small build size [16]. In addition, they are impractical to implement as the type of usable material is small and the output must be processed again after printing. FDM is the most widely used technology; in this technique, a thermoplastic resin is heated in liquid form, extruded through a nozzle, and cooled on a flatbed to deposit it. The advantage of the use of a filament in FDM is that it is inexpensive and can be used for materials with various properties. In general, the greatest advantage of FDM is that it is possible to control the packing density of the output so that low-density tissues, such as lungs, can be modeled [17,18]. Studies to evaluate the properties of various materials by analyzing the tissue equivalent and radiological density using the FDM and SLA methods have been conducted [19]. In addition, various metallic and non-metallic 3D printing materials have been printed and comparatively analyzed using various energies, such as kilovoltage computed tomography (kVCT) and megavoltage computed tomography (MVCT) [20]. kVCT is used in clinical practice and can use kilovoltage energy to analyze all the organs in a patient's body, including lungs, dense bone structures, and tissues with densities similar to water. This yields the same results for the powder and resin used in SLS and SLA printers, and notably, both printers have some limitations in terms of low densities [21].

However, materials containing high-density metals related to bone showed inhomogeneity owing to high artifacts in the commonly used kVCT [22]. MVCT imaging can use energies in the megavoltage range to more effectively reduce artifacts for dense-metal-using energies (~3.5 MeV) in the treatment area where the Compton effect is dominant, but the high energy makes it difficult to use in the context of clinical practice [23].

Although various studies have demonstrated the radiographic characterization of 3D printing materials, materials with high density such as bone should also be verified [24]. Therefore, research on human-like materials is important in order to ensure that more human-like phantoms can be printed [25].

In this study, a phantom for electron density insertion was created to analyze a material with a density similar to that of the human body using an FDM 3D printer. Additionally, the uniformity of the 3D printing materials was analyzed using a clinical linear accelerator. The study focused on evaluating the various parameters that affect print quality and materials with similar densities to the commercially available electronic density phantoms.

## 2. Materials and Methods

### 2.1. 3D Printer

The printer used was a Raise3D pro2 plus model (Santa Ana, CA, USA) with the FDM method. It is a precision 3D printer with a maximum size of 305 mm in diameter and 605 mm in height, and the extruder's X/Y resolution is ±0.781 μm while the Z resolution is ±0.0781 μm. Various layer thicknesses can be printed by controlling various nozzles with a size of 0.2–0.8 mm; in this study, a 0.4 mm nozzle was used for printing. A stereolithographic-generated file with information about the volume of the output in 3D printing was created using the Inventor 2021 program. Subsequently, it was output to a 3D printer using ideaMaker (Raise3d software) Version 4.0.1, a software program that

performs conversion to the G-Code format with conditions (speed, temperature, position, infill density, etc.) that can be implemented on a 3D printer.

### 2.2. Infill Density for the 3D Printing Materials

Infill density is a key parameter of FDM 3D printing that controls the internal mass density of an object; it indicates a space filling of between 10% and 100% as a ratio of the volume of air to the volume of the filament material. A filament material was used as the printing material; therefore, a thermoplastic resin that can be used within the extruder temperature range was used and non-metallic filaments and metal filaments were used as commercially available filaments. Non-metallic filaments with a density of 1.04–1.25, polylactic acid (PLA), acrylonitrile butadiene styrene (ABS), and thermoplastic polyurethane (TPU) were used, and steel and copper were used as metallic filaments. The materials used are listed in Table 1. The filaments have a diameter of 1.75 mm and readily available filaments from major manufacturers were used. The extruder temperature of the above materials is in the range of 190–250 °C, and the output characteristics of each material are different; however, they are printed using the same process, and the temperature of the extruder during printing was maintained using the printer's control board. The temperature of the heating plate was set at 60–75 °C, except for ABS. The recommended temperature for ABS was set to 110 °C, owing to the characteristics of the material. The temperature of the heating plate is important as it helps to achieve sufficient adhesion between the material and the aluminum plate, as well as to prevent curling, which commonly occurs in high-temperature 3D printers.

**Table 1.** Filaments used in the FDM printer.

| Filament Name | Manufacturer | Extruder Temperature (°C) | Physical Density (g/cm$^3$) |
| --- | --- | --- | --- |
| PLA | Basf Ultrafuse | 205 | 1.25 |
| ABS_Fusion+ | Basf Ultrafuse | 250 | 1.075 |
| TPU 85A | Basf Ultrafuse | 200 | 1.082 |
| Steel | ColorFabb | 210 | 3.13 |
| Copper | ColorFabb | 220 | 3.8 |

### 2.3. Fabrication of PLA/PBAT/Baso$_4$ Mixed Filaments

PLA, polybutylene adipate terephthalate (PBAT), and Baritop HD® barium powder (BaSO$_4$, 99% w/w barium sulfate; Kaigen Pharram Co., Ltd., Osaka, Japan) were mixed to print the bone density. Baritop HD powder, which is barium sulfate for medical use, is BaSO$_4$ from which heavy metals have been removed, which is harmless to the human body and is used as an angiogenesis agent in hospitals. First, PLA and PBAT pellets were stirred at a temperature of 250 °C for 60 min. After both materials had liquefied during the stirring process, they were mixed by rotating the double blade inside. PLA, which is the base of the filament, is widely used as a biodegradable resin and blended with PBAT to offset the disadvantage of poor flexibility. Subsequently, BaSO$_4$ powder was added and further mixed at a temperature of 250 °C for 120 min. Alcohol (83%) was then added to melt the PLA/PBAT surface so that the BaSO$_4$ adhered well. The alcohol evaporated during synthesis. Finally, the mixed composite was cut into small pieces and dried at room temperature. The synthesized composite filament was manufactured using the 3DK SJ25 extrusion line with an extrusion nozzle of 1.75 mm, as shown in Figure 1. The extruder also extruded at a constant screw speed and at 250 °C, and the extruded filaments were cooled using two types of water baths instead of air cooling. Thermal deformation was minimized by following the hot water bath line with a temperature of 50 °C and a warm water bath of 20 °C. After drying the wet filament using a dryer, it was confirmed, using a laser, that the filament was manufactured to a thickness of 1.75 mm for thickness measurement. The final manufactured filament was wound using a winding machine.

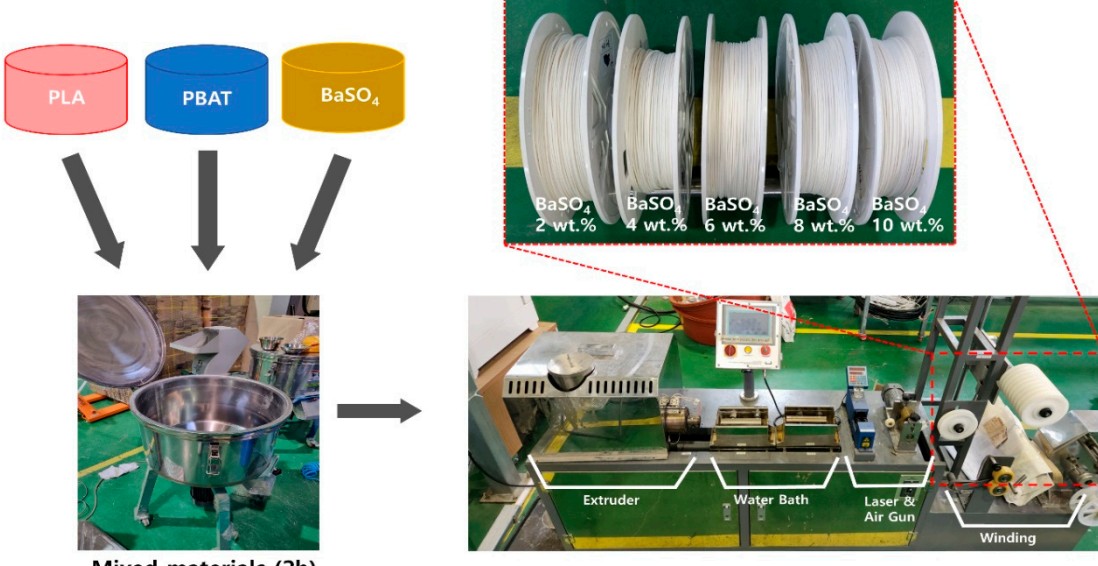

**Figure 1.** Filament manufacturing process and schematic diagram of the manufactured PLA/PBAT+BaSo₄ mix filament.

### 2.4. CT Scan

The electron density of the fabricated filament insert was measured using the Multi-plug CIRS® phantom 062 (Computerized Imaging Reference Systems, Norfolk, VA, USA) [26]. The CT image was scanned using GE Light Speed 16 (GE Medical Systems, Milwaukee, USA) equipment, and a peak tube potential of 120 kilovoltage peak (kVp) with an exposure of 200 mA was used. The thickness of the CT unit, 1.25 mm slice thickness, was selected. Next, the scanned CT image was imported into Eclipse contouring software to obtain the Hounsfield unit (HU) value for the image of the insert, and the average value of the CT number was measured by designating three regions of interest (ROIs) in the obtained image [27,28]. The CT image capture and analysis of the fabricated material are shown in Figure 2.

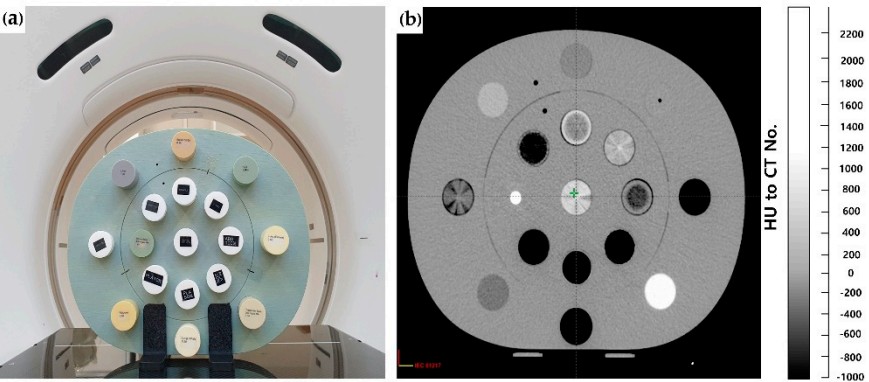

**Figure 2.** Electron density analysis of 3D printing materials using kVCT. (**a**) 3D printing material (white insert with black label) produced in this study and insert material with well-known electron density from CIRS. (**b**) Phantom image with electron density obtained using kVCT (0 HU: water density, negative HU: material with less density than water, positive HU: material with more density than water).

### 2.5. Uniformity

A square-shaped test block was printed to measure and analyze the electron density uniformity of the printed materials using the high energy photon beam from Clinac iX

(USA, Varian), a linear accelerator used for treatment. To achieve this, various 3D materials were printed and compared with a solid water phantom (PTW Freiburg, Germany). The experimental arrangement is shown in Figure 3. A Gafchromic EBT3 film (International Specialty Products Ashland Inc., Covington, KY) was placed under the printout. A layer of solid water (Gammex-RMI, Middleton, WI) was placed underneath the test block to provide sufficient accumulation and scattering. Directional labels were assigned to the test blocks for reproducibility and alignment between the beam irradiation plan and actual irradiation. The images were obtained after alignment using kVCT.

As shown in Figure 3, the phantom was irradiated with a radiation dose of 300 cGy using a 6 MV photon beam with a field size of 10 cm x 10 cm at source-to-surface distance (SSD) 100 cm. Doselab Pro 6.80 Version (Mobius Medical Systems, Houston, TX, USA) was used to analyze the dose distribution of the radiation irradiated through the solid water layer and the EBT3 film positioned under the 3D printing material.

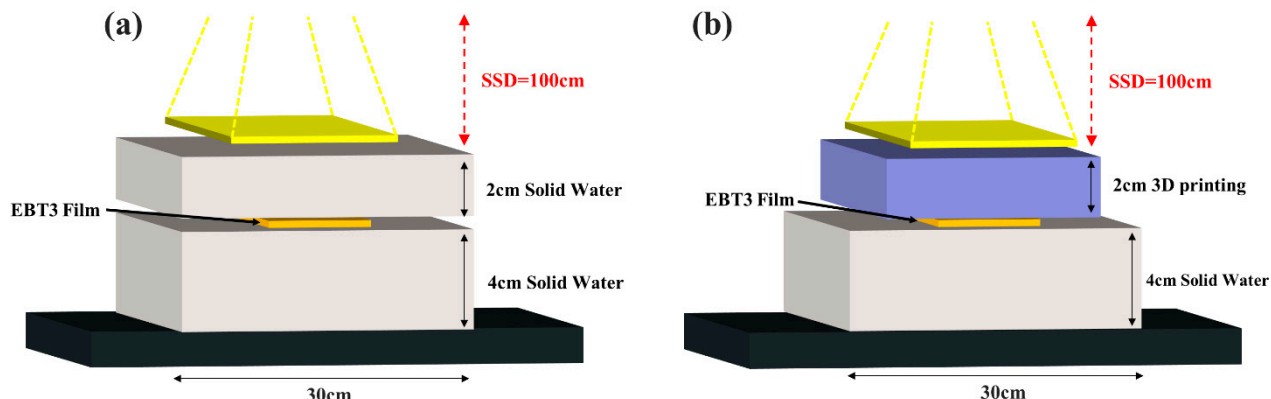

**Figure 3.** Irradiation of a clinical photon beam using a linear accelerator. (**a**) Photon beam irradiating the EBT3 film (PTW solid water layer with a thickness of 2 cm below). (**b**) Photon beam irradiating the EBT3 film (3D printed layer with a thickness of 2 cm below). Experiment consisting of inserting an EBT film between phantoms to evaluate the density uniformity of the phantom using a 6 MV photon beam. (**a**) Only commercially available solid water phantoms from PTW were used. (**b**) Solid water phantoms from PTW and 3D printed phantoms were used as substitutes in the experiment.

## 3. Results

### 3.1. Comparison of the HU Value for Infill Density and Infill Patterns

CT images were transferred to the Eclipse treatment planning system to compare the HU values for 3D printed materials. The standard deviation of the average HU value of the ROI is shown in the CT voxel data, and the size of the ROI was measured to be 1.75 cm × 1.75 cm, which is half the size of the internal insert. Figure 4 shows the results of comparing the 3D-printed materials, and the infill density was demonstrated by printing from 10% to 100%.

As shown in Figure 4, it can be seen that the values of HU are different according to the infill density of PLA, ABS, TPU, steel, and copper. Since the density of PLA, ABS, and TPU is close to 1 ($g/cm^3$), it was confirmed that the lower the infill density, the lower the HU value. On the other hand, since steel and copper are metallic materials, they have very high HU values compared to other materials even if the infill density is lowered. Among the filament materials, PLA had the highest HU value ($110 \pm 20$) and TPU had the lowest HU value ($-113 \pm 12$). It can be observed from Table 1 that ABS has the lowest physical density, but the HU value measured after outputting the actual ABS material is ($0 \pm 20$), which is a higher value. This is due to the effect of the ABS material on heat during the heat output process and the output characteristics due to heat shrinkage. The HU values of steel and copper, which are metal filament materials, can be obtained by printing from 30% to 100% of the infill density. At a low content of less than 30%, the inserts for low

filling output in the extrusion process of 3D printing filaments were not manufactured. In the case of steel and copper, the HU value with 30% infill is close to dense bone (about 800 HU); however, it is difficult to confirm the uncertainty of the material in the kVCT image acquired at 120 kVp owing to the artifact effect that occurs in the metal material [21].

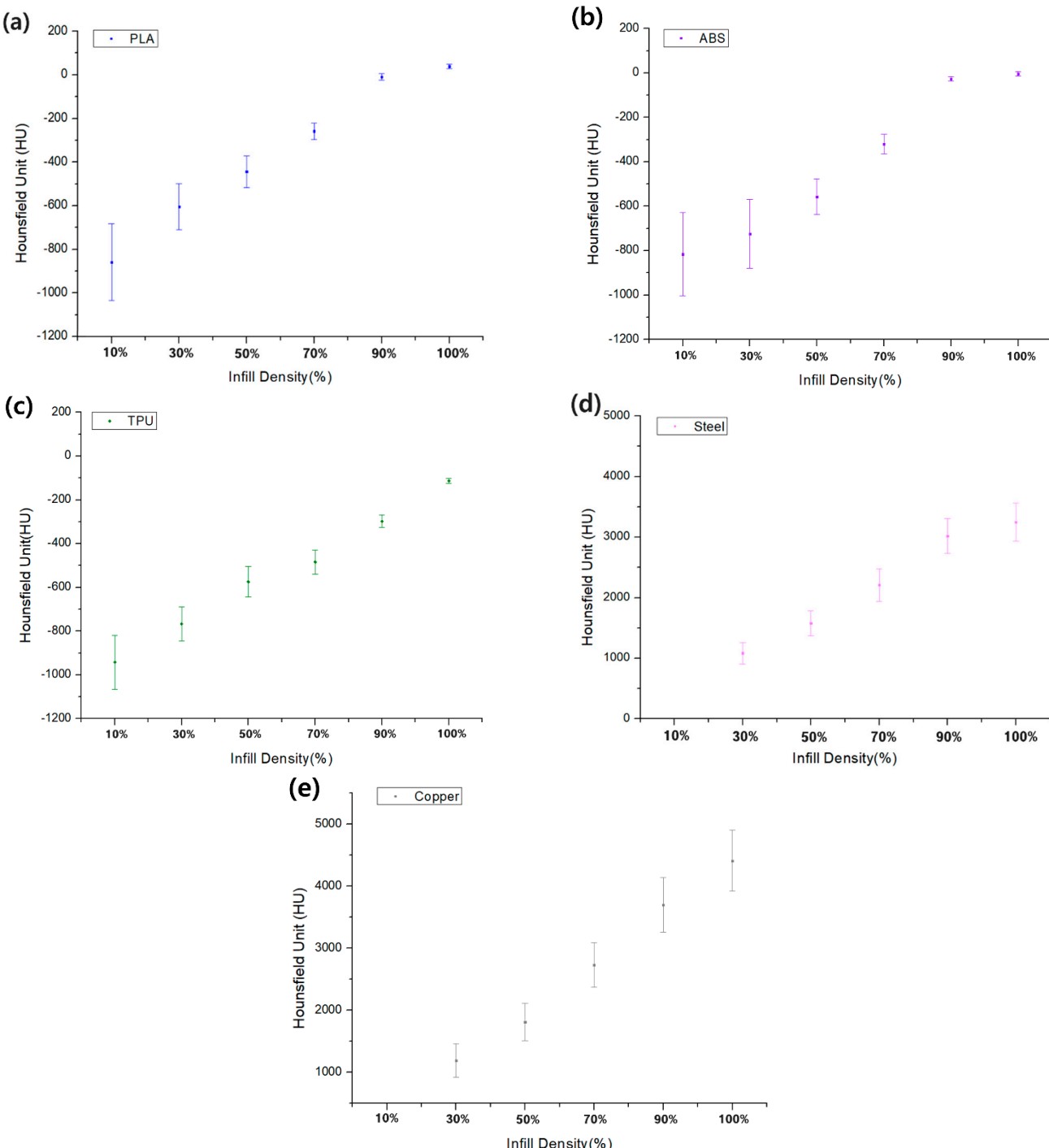

**Figure 4.** Comparison of Filament HU Values and Infill Density: (**a**) PLA, (**b**) ABS, (**c**) TPU, (**d**) Steel, (**e**) Copper.

### 3.2. Characteristics of Mixed Filament

A characteristic evaluation was performed after mixing $BaSO_4$ and PLA/PBAT in five ratios. BaSO4 was synthesized in five ratios of 2%, 4%, 6%, 8%, and 10% based on

PLA/PBAT, as listed in Table 2. The comparison is shown in Figure 5. The results were obtained through the same analytical method as the commercial filament materials. The material with the most similar electron density to trabeculae bone was obtained when $BaSO_4$ was mixed at a ratio of 8%, and the HU value was $905 \pm 7$.

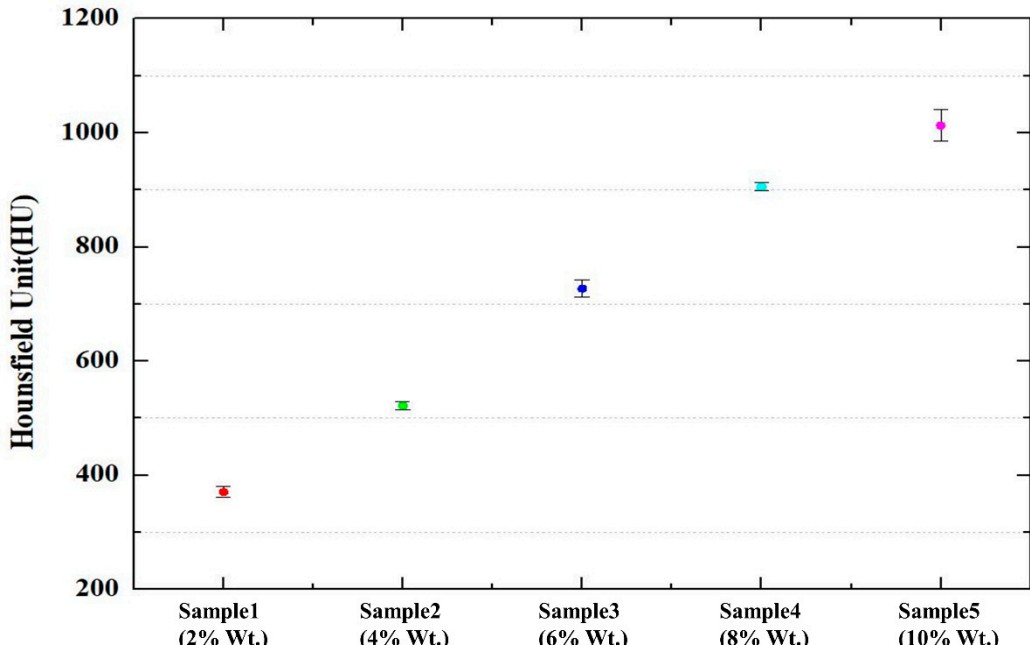

**Figure 5.** Comparison of HU values at wt. ratios from 2% to 10% for fabricated PLA/PBAT/$BaSO_4$ materials.

**Table 2.** Comparison of HU values based on the produced PLA+PBAT+BaSO4 mixed filament ratio.

| $BaSO_4$ Ratio (%) | PBAT Ratio (%) | PLA Ratio (%) | HU Value |
|---|---|---|---|
| 2 | 40 | 58 | $371 \pm 9$ |
| 4 | 40 | 56 | $522 \pm 7$ |
| 6 | 40 | 54 | $727 \pm 15$ |
| 8 | 40 | 52 | $905 \pm 7$ |
| 10 | 40 | 50 | $1013 \pm 28$ |

*3.3. Analysis of Radiation Effect in Clinical Photon Beam Using Gafchromic Film*

A grid pattern appeared clearly for the objects with variable density through infill density as a result of the image measurement of a 3D-printed rectangular phantom using CT. The HU value was the same as in the CT-inserted phantom, which can be explained by the air gap characteristics of the variable density for the low-density phantom [5]. The depth of the dose plane for the 6 MV beam was the same, and we marked the position of the coordinates on the edge of the phantom for each output to reduce the positional error. The result of the beam profile measured using the EBT3 film is shown in Figure 6. The comparison shows that the beam profile is distributed within 3% of all three types of filaments (PLA, ABS, and TPU) with infill densities of 10%, 30%, 50%, 70%, 90%, and 100%.

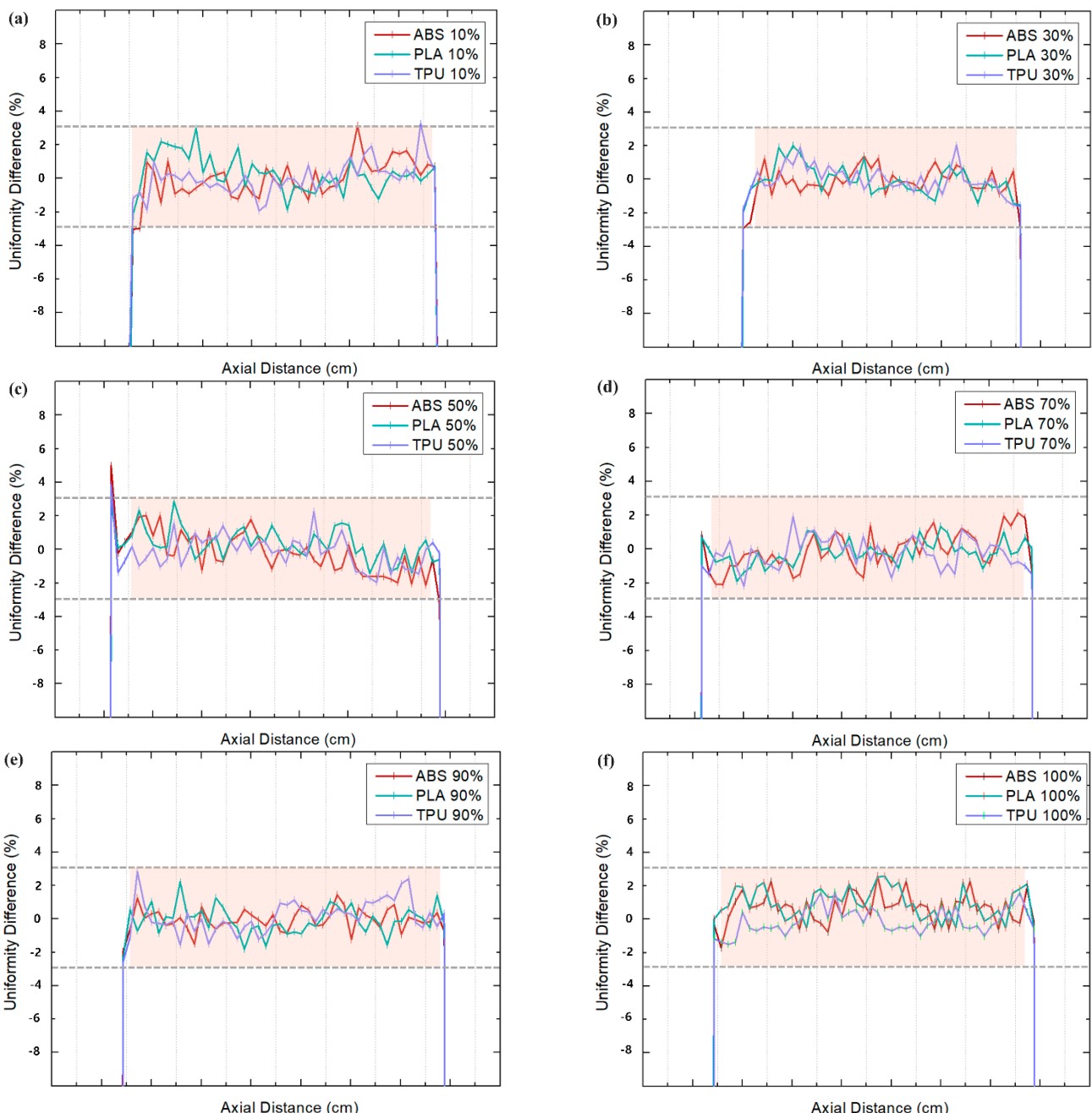

**Figure 6.** Comparison of uniformity of 3D printing materials (ABS, PLA, TPU) for (**a**) infill density 10%, (**b**) infill density 30%, (**c**) infill density 50%, (**d**) infill density 70%, (**e**) infill density 90%, (**f**) infill density 100%.

## 4. Discussion

We developed the FDM 3D printing filament material with similar electron density to real human organs and bones and analyzed its characteristics in this study. As shown in Table 3, by controlling the infill density of commercially produced filament materials, it was possible to create a 3D printing material corresponding to the electron density of a real human organ. As shown in Table 3, by controlling the infill density of commercially produced filament materials, it was possible to create a 3D printing material corresponding to the electron density of a real human organ. In addition, as shown in Table 2, by varying the ratios of BaSO4, PBAT, and PLA, bone filament materials with various electron densities, from trabecular bone to dense bone, were developed. Dancewicz et al. conducted a study with an electron density similar to that of human tissue using ABS, PLA, WoodFill,

CopperFill, and BronzeFill [20]. The infill density of ABS and PLA was adjusted to find the density corresponding to human organs such as lung, liver, and adipose, and the results were similar to those of this study. On the other hand, in their study, the bone material was manufactured using commercially available metal materials, and the electron density was 2–3 times higher than that of actual human bone. Kairn et al. used stone filament to develop a filament material with a density corresponding to that of bone, but the uniformity was low [29]. On the other hand, we were able to produce a filament with an electron density very similar to that of human bone, as shown in Table 3, by mixing a non-metal material rather than a metallic material. The use of 3D printing with the bone density obtained in our study can provide a more uniform and high-quality bone density.

　　As a result of our comparative analysis of the infill density of 3D printed materials, a larger standard deviation was observed in the region with a lower internal filling value, and a low-density volume was achieved by mixing large air voxels with PLA filament voxels [30]. The error bar in the graph shown in Figure 4 indicates that it has a low error value at a relative density of 70% or more. Madamesila et al. also recommended 60% more infill density based on an analysis of the electron density of polystyrene filaments, so that the deviation value of the material may be improved through mutual comparison [18]. This shows that the use of the DQA phantoms in the low-density range is limited, and the accuracy of the dose calculation seems limited in terms of dosimetry and quality assurance. Furthermore, when irradiating low-density phantoms from all directions rather than a single beam in one direction, the voids between multiple layers cannot be reliably reproduced when a 3D printed phantom with an infill ratio of less than 70% is used. Phantom dosimetry is very sensitive to the settings of a small stereotactic irradiation surface with dimensions similar to the air gap; therefore, it is difficult to guarantee the quality of dosimetry in the case of a very low-density 3D manufactured phantom [31].

　　Filaments mixed with a metal exhibit other effects, such as a photoelectric effect at kV energy, which affects the linearity of the curve and results in a high deviation. It was also demonstrated in the studies of Dancewicz et al. and Ma et al. that the CT values of the metallic objects themselves were erroneously expressed in the kVCT image [20,21]. In the kilovoltage peak images, which are energies generally used for patients, it can be observed that the CT values of metallic materials exhibit significantly high off-axis fluctuations and errors in the linear graph. Therefore, such CT values show that high-density inserts and implants such as metals in 3D printed materials should not be used for DQA due to the artifact effect.

**Table 3.** Comparison of CIRS phantom tissue equivalent materials and 3D printing materials.

| Tissue | CIRS Material (HU) | Std. Dev | Filament | 3D Printing Materials (HU) | Std. Dev. |
|---|---|---|---|---|---|
| Water | 3.0 | 1.6 | ABS (100%) | −2 | 6 |
| Adipose | −63.3 | 2.9 | PLA (90%) | −60 | 8.3 |
| Breast | −35.1 | 2.5 | TPU (100%) | −32 | 8 |
| Liver | 47.1 | 2.0 | PLA (100%) | 42 | 8 |
| Lung inhale | −772.1 | 7.5 | TPU (30%) | −766 | 84 |
| Lung exhale | −492.1 | 6.1 | PLA (50%) | −490 | 53 |
| | | | TPU (70%) | −484 | 41 |
| Trabecular bone | 236.9 | 7.5 | $BaSO_4$ Mix (2%) | 371 | 9 |
| Dense bone | 898.8 | 28.4 | $BaSO_4$ Mix (8%) | 905 | 7 |

　　The uniformity was confirmed based on the electron density of the fabricated material, and it was confirmed that it can be applied to a linear accelerator beam for treatment. All the values obtained using a 6 MV photon beam are within a difference of 3%. Since a sufficient build-up region was formed in the slab phantom for the photon beam, the uniformity of the beam profile was within 3%. Since the lung phantom, which is a low-density area on CT, was fabricated using less than 70% of the infill density material, the air density increased compared to the material and the value was not uniform. However, it can be observed that

the uniformity of the profile was also within 3% in the photon beam, which is less affected by the air gap due to the characteristics of the high-energy photon beam. It is possible to experimentally measure the dose profile for the treatment beam when the output from the air cavity is used in photon beam therapy [32]. The dimensions obtained from the CT scan of the fabricated 3D printed material were reproducible within the plane and slice resolution of the CT scan we used. As a result, it was possible to manufacture and compare filament materials with similar densities to Gammax materials, and to verify human-like materials using them. Although it does not have a very low uncertainty as a solid water material does, it was able to produce a product with good performance, and the 3D printer provided the desired shape and usability.

## 5. Conclusions

In this study, we verified that the materials used for 3D printing have similar densities to those found in a real human body. We verified the effectiveness of using the human body equivalent material by comparing the filament material used in a 3D printer and commercial CT density phantom. The advantages of 3D printing are (1) low-cost technology and reduced cost of consumables, (2) flexibility of printing various shapes and relative electron densities, and (3) ease of operation. By developing this technology, it can be expected that the non-uniform model and characteristics of the human body can be modeled, and the quality of treatment can be guaranteed by making a 3D phantom based on actual patient information. Currently, research on the usefulness of 3D printing technology is being actively conducted in various fields of medicine and medical physics, and it is expected that the quality of treatment will be improved by patient-tailored DQA through this technology.

**Author Contributions:** Conceptualization, Y.C. and S.H.C.; methodology, S.H.C.; investigation, S.H.C. and K.B.K.; data curation, Y.C., Y.J.J. and K.B.K.; visualization, Y.C. and Y.J.J.; validation, Y.C., K.B.K. and J.B.; software, Y.J.J.; writing—original draft preparation, Y.C., Y.J.J., K.B.K. and S.H.C.; writing—review and editing, J.B. and S.H.C.; supervision, J.B.; project administration, S.H.C. All authors have read and agreed to the published version of the manuscript.

**Funding:** This research was supported by the Nuclear Safety Research Program through the Korea Foundation of Nuclear Safety (KoFONS) using the financial resource granted by the Nuclear Safety and Security Commission (NSSC) of the Republic of Korea (No. 2003013) & National Research Foundation of Korea (NRF) grant funded by the Korea government (Ministry of Sciences and ICT) (No. 2020M2D9A309417021).

**Institutional Review Board Statement:** Not applicable.

**Informed Consent Statement:** Not applicable.

**Data Availability Statement:** The data presented in this study are available upon request from the corresponding author. The data are not publicly available because the data also form part of an ongoing study.

**Conflicts of Interest:** The authors declare no potential conflicts of interest with respect to the authorship and/or publication of this article.

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
