# Peer review of "Characterization of Tissue Equivalent Materials Using 3D Printing for Patient-Specific DQA in Radiation Therapy"

_applsci, doi:10.3390/app12199768_

Round 1

Reviewer 1 Report

The article is good written, and the methods and the results have good presentation

but i have to ask for a major revision because the discussion should be re-written with the comparison with other similar studies

New references should be added to the discussion part

Author Response

Point 1: The article is good written, and the methods and the results have good presentation but i have to ask for a major revision because the discussion should be re-written with the comparison with other similar studies. New references should be added to the discussion part

 Response 1: Thank you for this suggestion and careful reading. We have applied that the discussion rewritten with the comparison with other similar studies and New references added to the discussion part.

Reviewer 2 Report

The paper entitled” Characterization of Tissue Equivalent Materials Using 3D 2 Printing for Patient Specific DQA in Radiation Therapy” studied the electron density of several commercial 3D printing materials and research materials for making phantom with bone-comparable properties. 3D printing is playing a more and more pivotal role in medical science and it is attracting enormous attentions. Generally, the current manuscript is easy to read through with logic being fine. However, there are still some questions the authors should consider and address before the publication. I would recommend its publication in the journal of Applied Sciences if the authors address my concerns:

1.       The authors should briefly explain the two images in the figure 2 caption., particularly the left digital image, what are the colored chunks and what are the white chunks with black square in the center? Please label them in the left image and describe them in the caption. In the right image, please add a scale bar describing the range of density.

2.       The authors should claim more significance of this work. Is the electron density highly correlated to the physical density of the bone? Why does the electron density matter for the phantom materials? What about other mechanical properties of the materials?

3.       The authors used many abbreviation and terminologies that should be defined for the reading convenience of audience, such as kVCT and MVCT in the  introduction. Likewise, what are bone 200 and bone 800? Are they real bones or bone materials? What are their compositions?

4.       The authors prepared the composite materials PLA/PBAT/BaSO4. How is the homogeneity of the composite material? Is there any characterization related to the homogeneity, such as SEM image? Why did not the author study the uniformity/printability of this mixer?

5.       The HU values for different equivalent materials and 3D printing materials varies a lot, and they even have both positive and negative numbers. Does the polarity of HU value matter for a bone phantom?

6.       To make this manuscript more informative, some state-of-the-art works reporting 3D printing artificial bones/bone implants should be cited, such as: ACS Nano, 2021, 15, 14903-14914; Advanced Functional Materials 31.40 (2021): 2105190

7.       What are the HU values for real bones? There are mainly two kinds of bones, cortical bones and cancerous bones. What are the electron densities/HU values for these two bones?

Author Response

Point 1: The authors should briefly explain the two images in the figure 2 caption., particularly the left digital image, what are the colored chunks and what are the white chunks with black square in the center? Please label them in the left image and describe them in the caption. In the right image, please add a scale bar describing the range of density.

 Response 1: Thank you for this suggestion. The content of the two images in the caption of Figure 2 was explained and changed. “Figure 2. Electron density analysis of 3D printing materials using kVCT. (a) 3D printing material (white insert with black label) produced in this study and insert material with well-known electron density from CIRS, (b) Phantom image with electron density obtained using kVCT (0 HU: water density, negative HU: material less than water density, positive HU: material more than water density)” at line 161-165.

Point 2: The authors should claim more significance of this work. Is the electron density highly correlated to the physical density of the bone? Why does the electron density matter for the phantom materials? What about other mechanical properties of the materials?

Response 2: Thank you for careful reading. So, we claim more significance of this work that the electron density corrected to the physical density bone and the electron density matter for the phantom At line 37-42. “Since CT images used for treatment planning in radiation therapy are mainly ac-quired using energy of 50 keV or more, the interaction of radiation is mainly due to Compton scattering and is closely related to electron density for organs, bones and tumors. [8]. For human tissue materials, data on the electron beam attenuation coefficient can be represented by a single straight line, but for materials such as bone, the attenuation coeffi-cient is different [9].”

Point 3: The authors used many abbreviation and terminologies that should be defined for the reading convenience of audience, such as kVCT and MVCT in the introduction. Likewise, what are bone 200 and bone 800? Are they real bones or bone materials? What are their compositions?

Response 3: Thank you for this suggestion and careful reading. The abbreviation and terminologies has been defined for the convenience of audience. Rewritten as that sentence line 68-79. “In addition, various metallic and non-metallic 3D printing materials have been printed and comparatively analyzed using various energies such as kilovoltage computed to-mography (kVCT) and megavoltage computed tomography (MVCT) [20]. kVCT is used in clinical practice and can use kilovoltage energy to analyze all organs in a patient's body, lung, dense bone structures and tissues similar to water density. This yielded the same results for the powder and resin used in SLS and SLA printers, and in particular, both printers have some limitations at low density [21].

However, materials containing high-density metals related to bone showed inhomo-geneity owing to high artifacts in the commonly used kVCT [22]. MVCT imaging can use energies in the megavoltage range to more effectively reduce artifacts for dense metals us-ing energies (~3.5 MeV) in the treatment area where the Compton effect is dominant, but the high energy makes it difficult to use in clinical practice [23].”

Also, Bone 200 is the name of the bone insert phantom with 200HU value, and Bone 800 is the name of the bone insert phantom with 800HU value. For better understanding, the name has been changed to correspond to the electron density.  Bone 200 -> Trabecular bone  , Bone 800 -> Dense Bone.

Thanks to the reviewer's observations, we were able to change the name to make it easier for audiences to understand.

Point 4: The authors prepared the composite materials PLA/PBAT/BaSO4. How is the homogeneity of the composite material? Is there any characterization related to the homogeneity, such as SEM image? Why did not the author study the uniformity/printability of this mixer?

Response 4: Thank you for this suggestion. In this study, a 3D printing material was developed for application to radiation therapy DQA. Therefore, as in Section 2.5, the uniformity of the composite material was analyzed by evaluating the homogeneity of electron density in radiation. We rewrite it as line 168-170. “A square-shaped test block was printed to measure and analyze the electron density uniformity of printed materials using the high energy photon beam from Clinac iX (USA, Varian), a linear accelerator used for treatment.”

The uniformity can be confirmed by standard deviation as shown in Table 2.

Point 5: The HU values for different equivalent materials and 3D printing materials varies a lot, and they even have both positive and negative numbers. Does the polarity of HU value matter for a bone phantom?

Response 5: Thank you for this suggestions. It has been modified as line 161-165. “Electron density analysis of 3D printed materials using kVCT. (a) 3D printing material (white insert with black label) produced in this study and insert material with well-known density from CIRS, (b) Phantom image with electron density obtained using kVCT (0 HU: water density, negative HU: material less than water density, positive HU: material more than water density)”

Point 6: To make this manuscript more informative, some state-of-the-art works reporting 3D printing artificial bones/bone implants should be cited, such as: ACS Nano, 2021, 15, 14903-14914; Advanced Functional Materials 31.40 (2021): 2105190

Response 6: Thanks for providing a good reference. It was applied as a reference paper for this manuscript, and it is shown in line 83.

Point 7: What are the HU values for real bones? There are mainly two kinds of bones, cortical bones and cancerous bones. What are the electron densities/HU values for these two bones?

Response 7: Thank you for this suggestion and careful reading. We modified it as follows: the bone 200 and bone 800 materials indicated in point 3 are the standard HU values for cortical bone (dense bone) and cancerous bone (trabecular bone), and are shown in Table 3.

The cancerous bone(Trabecular bone) HU value : 236.9 , cortical bone(Dense bone) HU value : 898.8

We believe that these modifications have strengthened the manuscript and hope that the revised manuscript is suitable for publication in Applied Science.

Sincerely,

Sang Hyoun Choi

Reviewer 3 Report

The manuscript entitled “Characterization of Tissue Equivalent Materials Using 3D 2 Printing for Patient Specific DQA in Radiation Therapy” focuses on a very interesting and actual subject. However, the manuscript in not clear neither precise concerning important definitions, experimental aspects, results discussion and language. It needs a deep revision. In this way, I recommend the manuscript re-submission after structural and scientific revision.   

Some more detailed comments:

- Abstract can be improved, eg: there is no indication of the PBAT/PLA percentages studied.

- There should be a space between the definition of a quantity or technique and the abbreviation. The same for a quantity and the presentation of the respective unit (apply to the entire document).

- The language is not clear in all sections of the manuscript. The presentation of some concepts is confuse, for instance, line 40-41: “…, although standard values are provided for each diseased material”, diseased material? What this really means? Biological tissues? Different tissue equivalent materials since there are different biological tissues (basically different type of phantoms depending on the diseased organ and tissue). It must be re-written in a clear and precise way in the entire document. (At line 69-70 there is a similar situation, it is bone or bone phantom? And so on…)

 - Line 37: QA or DQA?

 - I suggest present the definition of kVTC and MVTC to turn the reading easier.

 - Line 80: “…The 3D printer used the pro2 plus model of Raise3D…” – Rewrite please (ex: The printer used was a Raise3D pro2 plus model).

 - Line 92: “…Infill density is a key parameter in the printer software…” – It is true, but it is not just an important parameter in the software, it is a key parameter of 3D printing! Rewrite please.

- Line 97-99: Authors mention non-metallic filaments but then say “as well as steel…” – the problem is the English, please rewrite.

- The table 1 should include the temperature of the extruder (and it would be visuality more attractive if centered).

- At 2.3 section it must be indicate the percentage of the components in the mixture and the temperature at which they were mixed. Also, it is not clear, PLA was used as powder or as filament? And alcohol was added to melt? Please clarify.

- Please present the definition of kVp, HU, etc, they are very specific and present their definition will make the reading easier for a non-expert in the subject.

- Section 3: Uniformity – Uniformity of what? Once again is not presented in a clear and precise way. Uniformity of the material produced in terms of dose uniformity? In terms of reproducibility of the biological tissues? etc...

- “… the radiation beam was irradiated for materials printed” – does not make sense the radiation beam be irradiated! English, please rewrite.

- Line 161: “… A plan was drawn up to irradiate a clinical photon beam of…”: again, it is the irradiation of a beam? Or the irradiation of a phantom? Please clarify. Also correct figure 3 sub-tittle.

- Line 163: delivered to or delivered by a linear accelerator?

- Figure 4 is not clear. The use of figures with identical scales would facilitate the interpretation of the data. Also, the discussion about it and the values presented are not in agreement with the graphs.

- The entire Discussion section needs to be re-written. The main problem is the lack or precision. For instance: “The purpose of this study is to analyze and recommend materials that can be used in FDM-type 3D printers to provide accurate human-like density.” The goal of the study was to identify the human-like density or to develop materials with similar density to real human body?

etc...

Author Response

Point 1: Abstract can be improved, eg: there is no indication of the PBAT/PLA percentages studied.

Response 1: Thanks for this suggestion. We improved the abstract and Table 2 shows the PBAT/PLA percentage indication of our study.

Point 2: There should be a space between the definition of a quantity or technique and the abbreviation. The same for a quantity and the presentation of the respective unit (apply to the entire document).

Response 2: Thank you for this suggestion and careful reading. We have applied the same for the quantity and indication of the unit in question with the entire document.

Point 3:  The language is not clear in all sections of the manuscript. The presentation of some concepts is confuse, for instance, line 40-41: “…, although standard values are provided for each diseased material”, diseased material? What this really means? Biological tissues? Different tissue equivalent materials since there are different biological tissues (basically different type of phantoms depending on the diseased organ and tissue). It must be re-written in a clear and precise way in the entire document. (At line 69-70 there is a similar situation, it is bone or bone phantom? And so on…)

Response 3: I fully agree with the reviewer's comments. The entire document has been rewritten clearly and precisely.

We changed the contents of lines 45-48 to “although phantom manufacturer provided standard values for several reference material, research on materials with various densities for each organ is needed considering the dif-ferent characteristics of each individual patient [10].”

And line 80-83 has rewritten  “Although various studies have demonstrated the radiographic characterization of 3D printing materials, materials with high density such as bone should also be verified [24]. Therefore, research on human-like materials is important to ensure that more human-like phantoms can be printed [25].” Thank you for your careful reading.

Point 4: Line 37: QA or DQA?

Response 4: Thank you for this suggestion. We corrected it with Delivery QA(DQA).

Point 5: I suggest present the definition of kVTC and MVTC to turn the reading easier.

Response 5: Thank you for this suggestion. Line 69-70 present the definition of kVCT and MVCT .

Point 6: Line 80: “…The 3D printer used the pro2 plus model of Raise3D…” – Rewrite please (ex: The printer used was a Raise3D pro2 plus model).

Response 6: Thank you for this suggestion. Rewritten as that sentence Line 91 “The printer used was a Raise3D pro2 plus model”.

Point 7: Line 92: “…Infill density is a key parameter in the printer software…” – It is true, but it is not just an important parameter in the software, it is a key parameter of 3D printing! Rewrite please.

Response 7: Thank you for this suggestion. Rewritten as that sentence Line 103 “Infill density is a key parameter of FDM 3D printing”

Point 8: Line 97-99: Authors mention non-metallic filaments but then say “as well as steel…” – the problem is the English, please rewrite.

Response 8: The sentence was changed through the correct point of view of the reviewer. Line 108-111 ”Non-metallic filaments with a density of 1.04 to 1.25, polylactic acid (PLA), acrylonitrile butadiene styrene (ABS), and thermoplastic polyurethane (TPU), were used, and steel and copper were used as metallic filaments. The materials used are listed in the Table 1.”

Point 9: The table 1 should include the temperature of the extruder (and it would be visuality more attractive if centered).

Response 9: Thank you for your careful reading. We have included the extruder temperatures in Table 1. Thanks to reviewers, it makes Table 1 more visually appealing.

Point 10: At 2.3 section it must be indicate the percentage of the components in the mixture and the temperature at which they were mixed. Also, it is not clear, PLA was used as powder or as filament? And alcohol was added to melt? Please clarify.

Response 10: Thank you for this suggestion. We have indicated the percentage of the components in the mixture and the temperature At 2.3 section. Also, We indicated that PLA/PBAT was used as the pellet type. And we mentioned the addition of alcohol to dissolve both pellets in the text.

Point 11: Please present the definition of kVp, HU, etc, they are very specific and present their definition will make the reading easier for a non-expert in the subject.

Response 11: Thank you for your careful reading. We have included the definition of kilovoltage peak (kVp) at line 153 and the definition of hounsfield unit (HU) at line 156.

Point 12: Section 3: Uniformity – Uniformity of what? Once again is not presented in a clear and precise way. Uniformity of the material produced in terms of dose uniformity? In terms of reproducibility of the biological tissues? etc...

Response 12: Thank you for this suggestion. It means that the material produced in terms of dose uniformity. We rewrote section 2.5.

Point 13: “… the radiation beam was irradiated for materials printed” – does not make sense the radiation beam be irradiated! English, please rewrite.

Response 13: Thank you for your careful reading. We would have made a huge mistake if the reviewer didn't mention it. We rewrote that sentence “As shown in Fig. 3, the phantom was irradiated with a radiation dose of 300 cGy using a 6 MV photon beam with a field size of 10 cm x 10 cm in source at surface distance (SSD) 100 cm.” at line 178-180.

Point 14: Line 161: “… A plan was drawn up to irradiate a clinical photon beam of…”: again, it is the irradiation of a beam? Or the irradiation of a phantom? Please clarify. Also correct figure 3 sub-tittle.

Response 14: Thank you for this suggestion. We have rewritten clarify “As shown in Fig. 3, the phantom was irradiated with a radiation dose of 300 cGy using a 6 MV photon beam with a field size of 10 cm x 10 cm at source to surface distance (SSD) 100 cm. Doselab Pro 6.80 Version (Mobius Medical Systems, Houston, USA) was used to ana-lyze the dose distribution of the radiation irradiated through the solid water layer and the EBT3 film positioned under the 3D printing material.” at line 178-182.

And corrected the sub-title of figure 3 on lines 184-189.

Point 15: Line 163: delivered to or delivered by a linear accelerator?

Response 15: Thank you for this suggestion. We have rewrote clearly “the radiation irradiated through the solid water layer and the EBT3 film positioned under the 3D printing material. “ at line 181-182.

Point 16: Figure 4 is not clear. The use of figures with identical scales would facilitate the interpretation of the data. Also, the discussion about it and the values presented are not in agreement with the graphs.

Response 16: Thank you for this suggestion. In Fig. 4, (a), (b), and (c) show the same scale for non-metallic filaments. However, since (d) and (e) are metallic filaments, high-scale values came out, and according to this reviewer's suggestion, we have rewritten the scales of (d) and (e).

And the discussion and suggested values were corrected.

Point 17: The entire Discussion section needs to be re-written. The main problem is the lack or precision. For instance: “The purpose of this study is to analyze and recommend materials that can be used in FDM-type 3D printers to provide accurate human-like density.” The goal of the study was to identify the human-like density or to develop materials with similar density to real human body?

Response 17: Thank you for this suggestion. We have rewritten the discussion section and we applied that sentence “The purpose of this study is to develop and analyze the materials with similar density to real human tissue-equivalent materials”

We believe that these modifications have strengthened the manuscript and hope that the revised manuscript is suitable for publication in Applied Science.

Sincerely,

Sang Hyoun Choi

Reviewer 4 Report

This manuscript "Characterization of Tissue Equivalent Materials Using 3D 2 Printing for Patient Specific DQA in Radiation Therapy"presented an interesting study, is well written, properly organized and fits within the scope of the journal . I my opinion the work if correctly carried out and have some novelty. I suggest accept in the present form. 

Author Response

Point 1: This manuscript "Characterization of Tissue Equivalent Materials Using 3D 2 Printing for Patient Specific DQA in Radiation Therapy"presented an interesting study, is well written, properly organized and fits within the scope of the journal . I my opinion the work if correctly carried out and have some novelty. I suggest accept in the present form.

Response 1: Thank you for your suggestion and a good look at our article. We will do more advanced 3D printing research including this paper. Thanks again to the reviewer.

We believe that these modifications have strengthened the manuscript and hope that the revised manuscript is suitable for publication in Applied Science.

Sincerely,

Sang Hyoun Choi

Round 2

Reviewer 1 Report

good responses

Author Response

Thank you for your suggestion and a good look at our article. We will do more advanced 3D printing research including this paper. Thanks again to the reviewer.

Reviewer 3 Report

Thank you for having revised the manuscript accordingly reviewers’ suggestions. The revised version is much clearer and can now be accepted for publication after minor corrections.

Line 15: please replace “are used” by “were used”;

Table 1: Replace “Temperature” by “Extruder Temperature”; also delete an extra “)”;

Lines 143 and 147: insert a “of” after temperature (eg: “…temperature of 250 ºC).

Author Response

Thank you for this suggestion & careful reading. The manuscript was revised according to the reviewer's suggestion.

Point 1: Line 15: please replace “are used” by “were used”

Response 1: Thanks for this suggestion. Rewritten as that sentence Line 13 “were used”.

Point 2: Table 1: Replace “Temperature” by “Extruder Temperature”; also delete an extra “)”

Response 2: Thanks for this suggestion. Replaced as that Table 1 “Extruder Temperature” and delete an extra “)”.

Point 3: Lines 143 and 147: insert a “of” after temperature (eg: “…temperature of 250 ºC).

Response 3: Thanks for this suggestion and careful reading. Rewritten as that word “of” Line 131 “temperature of 250°C” and line 135 “temperature of 250°C”.

We believe that these modifications have strengthened the manuscript and hope that the revised manuscript is suitable for publication in Applied Science.

Sincerely,

Sang Hyoun Choi
